# A Multi-Sensor Based Roadheader Positioning Model and Arbitrary Tunnel Cross Section Automatic Cutting

**DOI:** 10.3390/s19224955

**Published:** 2019-11-14

**Authors:** Changqing Yan, Wenxiao Zhao, Xinming Lu

**Affiliations:** 1Department of Information Engineering, Shandong University of Science and Technology, Tai’an 271000, China; yancq@lreis.ac.cn; 2College of Computer Science and Engineering, Shandong University of Science and Technology & Research center of smart mine software engineering and technology, Qingdao 266590, China; wenxiao4629@163.com; 3Shandong Province Key Laboratory of Wisdom Mine Information Technology, Shandong University of Science and Technology, Qingdao 266590, China; 4Shandong Province Research Center of Intelligent Mine Software Engineering and Technology, Shandong Lionking Software Co., Ltd., Tai’an 271000, China

**Keywords:** automatic cutting, multi-sensor, roadheader, positioning model

## Abstract

Autonomous posture detection and self-localization of roadheaders is the key to automatic tunneling and roadheader robotization. In this paper, a multi-sensor based positioning method, involving an inertial system for altitude angles measurement, total station for coordinate measurement, and sensors for measuring the real-time length of the hydraulic cylinder is presented for roadheader position measurement and posture detection. Based on this method, a positioning model for roadheader and cutter positioning is developed. Additionally, flexible trajectory planning methods are provided for automatic cutting. Based on the positioning model and the trajectory planning methods, an automatic cutting procedure is proposed and applied in practical tunneling. The experimental results verify the high accuracy and efficiency of both the positioning method and the model. Furthermore, it is indicated that arbitrary shapes can be generated automatically and precisely according to the planned trajectory, employing the automatic cutting procedure. Therefore, unmanned tunneling can be realized by employing the proposed automatic cutting process.

## 1. Introduction

Tunnels are usually constructed for transportation in mountainous areas and for coal mining underground. The tunnel working surface is the most dangerous and labor intensive place, where accidents and fatalities occur at a rather high frequency. Therefore, unmanned work should be realized urgently to protect the lives and health of the miners and improve work efficiency. A boom-type roadheader is the principal device for coal mine tunneling. Its automation, intellectualization, and robotization are of great significance for realizing unmanned tunneling. To realize the automation and robotization of the roadheader, the positioning of the roadheader and the cutter should be addressed. However, due to high dense dust, vibration, low-illumination, complicated background, magnetic disturbance, and electrostatic interference in the underground environment the positioning problem is a very difficult one to solve. Additionally, the unavailability or disability of the signals that can be used on the ground increases the difficulty of underground roadheader positioning.

Over the last decades, significant research has been done on this topic [1,2,3,4,5,6]. With laser targets acting as information sources, a visual measurement system was presented for pose detection by utilizing the cross laser beams received [4]. While this system can be used to obtain a precise result for roadheader pose detection, this method is inevitably subject to accuracy loss due to the dense dust that visual methods may suffer from. A kinetic model was built to determine the pose of the roadheader [5]. Although the error of the model was reported to be 16 mm for practical tunneling, the core issue, how to measure the pose angles, was not clearly stated. An ultra-wide band pose detection system was designed for roadheader pose detection [2], which was shown to be able to meet the accuracy requirement of a fully mechanized excavation face. However, it is generally considered that it is very difficult to achieve an error of less than 10 cm due to the limitations of the ultra-wide band technique itself, as well as its increasing error with increasing distance. Furthermore, the deployment of this method is troublesome, and multiple ultra-wide band stations need to be deployed. A monocular vision and strap down inertial navigation based roadheader pose detection system was presented, and test results with an error of 1 cm could be achieved [6]. However, it is known that the inertial system is subject to accumulated error from double integration as well as drift error for long distance and extensive running time. A pose detection method based on laser cross was proposed [7], in which a fan-shaped laser beam between the transmitter and the receiver was employed to calculate the roadheader pose. Although this method was shown to exhibit good performance, it was subject to complicated deployment as well as error problems when employed in practice. An inertial sensors and a stereo camera system [8] was presented for positioning vehicles with high accuracy. Similar to the ultra-wide band method, there was difficulty with both deployment and with obtaining a high accuracy. An infrared LED based method was proposed to position the roadheader [9], which was reported to achieve an accuracy of less than 3 cm, but the infrared LED based method suffers from a similar problem to the ultra-wide band method, that is, the error increases with the increasing distance. Therefore, while the aforementioned methods are presented, to the best of our knowledge, none of them have wide-spread applications in a practical tunneling project. The reason for this is that these methods are either underdeveloped or are unable to adapt themselves for application in a complex underground environment due to their deficiencies. Dust, humidity, and complicated device deployment and operation for unskilled workers means that the methods mentioned may not be able to function properly on the ground. Further efforts need to be made to tackle the pose detection and localization problems of the roadheader. Generally, the commonly used positioning methods are inertial system based, laser device based, or ultra-wide band based. Of these, the inertial navigation and laser are two prevailing positioning methods. However, neither of them can be employed for high accuracy positioning in harsh underground environments independently. The laser device method, with the requirement of complicated deployment and operation, can be challenging in a complex environment such as underground mining. While the inertial system can be employed for positioning without external signals being required, as mentioned above, it suffers from an accumulated error for long duration and long distance employment. Nevertheless, an inertial system can yield the pose angles with a high precision due to the significantly reduced integration error compared to the error from double integration for the positioning [8]. While the laser based total station, a commonly used electronic/optical instrument for surveying, has the potential to achieve very high accurate positioning, the application of total station underground is very limited due to the requirement that it should be leveled before employment, which would be a rather difficult operation for unskilled workers with no special training in the aforementioned complex environment.

The critical data that need to be acquired for positioning are the posture angles of the roadheader and the coordinates of a fixed point on the roadheader when taking the roadheader as a rigid body [9]. To efficiently solve the positioning problem and realize high accuracy positioning, the inertial system based method and the total station based method should be combined. Although the inertial system should not be employed for precise localization due to the possible inertial accumulated error, it can be employed for generating the pose angles with a high accuracy. In this way, the error can be reduced significantly due to the fact that only single integration is involved compared to the double integration involved for direct localization. With the high accuracy pose angles, the coordinate system of the roadheader can be obtained. Furthermore, if the coordinate system of the total station can be measured, the exact position of the roadheader can be calculated without leveling the total station.

According to this idea, in this manuscript, a positioning method based on the combined total station and inertial navigation and the corresponding position model are presented for roadheader and the cutter positioning. Based on the positioning model and the developed trajectory generation method, an automatic cutting procedure is developed.

The following section details the methods for this research, including the positioning principle of the roadheader and the cutter, the cutter control, the repositioning method caused by the moving of the positioning device itself, as well as trajectory planning and the automatic cutting process. Practical experiments for the roadheader and cutter positioning and for the automatic cutting process are carried out, and the results are given in Section 3. In Section 4, discussions are made, and in Section 5 this paper is concluded.

## 2. Methods

### 2.1. Principle of Positioning and Cutter Control

For our positioning solution, multiple sensors or surveying instruments are required, including a north-seeking gyroscope, an inertial system, a total station, a target-ball, and sensors that measure the real-time length of the hydraulic cylinders governing the vertical and horizontal motion of the roadheader cutter. The total station is employed to track and measure the azimuth and inclination angles of the target-ball installed on the roadheader in real-time, from which the coordinates can be calculated. Generally, the total station is required to be leveled before surveying starts, which raises a great challenge for unskilled coal miners in an uneven underground environment. Measures should be found to avoid the need to level the total station. Therefore, a north-seeking gyroscope is employed to measure the pose angles’ yaw and pitch, and the roll of the total station, which will be taken into account for the positioning computing. Consequently, the bottom of the north-seeking gyroscope and the total station should be fitted and fixed on the same plane in an explosion proof box and deployed behind at a distance from the roadheader (Figure 1). The inertial system is used for the roadheader pose angles measurement, and should be installed in a similar way to the north-seeking gyroscope. The inertial system should be equipped with the marked direction in line with the roadheader’s forward direction. The inertial system can measure the precise pose angles in real-time so that the accurate pose angles can be obtained even when the roadheader is moving in a constantly vibrating state. The north-seeking gyroscope cannot measure the dynamically changing pose, so it is used to measure the pose of the total station with little vibration. When the position of the total station is changed, its pose angles should be re-measured. Since a roadheader can be taken as a rigid body [5], any point on the roadheader can be situated with the measured relative position regarding the trackball position and the trackball position from the total station. The cutting head can be positioned with a calculated fix point on the roadheader according to the kinematics of the roadheader, which will be clarified in the following section. According to the positioning principle, the arrangements of the positioning solution can be made in the underground worksite, and the positioning system is illustrated in Figure 1. In Figure 1, the cube with the vertices *f*1, *f*2, *f*3, *f*4, *f*5, *f*6, *f*7, and *f*8 represents the tunnel, the cube with the vertices *a*, *b*, *c*, *d*, *a*1, *b*1, *c*1, and *d*1 represents the roadheader body, *p* is the cutter, and *t* is the target-ball, which receives the laser from the positioning box adjacent to the face *f*1, *f*4, *f*5, *f*8.

#### 2.1.1. Positioning Workflow

In terms of the multiple sensors based positioning principle (Figure 1), a workflow is developed and is shown in Figure 2.

According to Figure 2, the total station will begin to search the target-ball after launch. Once the target-ball is found, the tracking mode will be started and the distance, azimuth, and inclination angles data will be output in real-time. Meanwhile, the north-seeking gyroscope outputs the yaw, pitch, and roll angles of the total station. The angles will be utilized for the total station coordinate system computing. With the total station coordinate system and the data from the total station, the target-ball coordinates can be calculated. Similarly, with the pose angles from the inertial system, the roadheader coordinate system can be calculated. As mentioned above, for any point on the roadheader, the relative coordinates with respect to the track-ball can be measured. With the relative coordinates and the roadheader coordinate system, any point on the roadheader can be positioned. To further calculate the cutter position, the rotation center on the roadheader should be calculated. Then, according to the kinematics of the roadheader [10], the cutter position relating to the rotation center can be calculated. In a similar way to the point positioning on the roadheader, the cutter position in the global coordinate system can be calculated. The positioning method is further detailed as follows.

To carry out the roadheader and cutter positioning, four coordinate systems are introduced, including the global coordinate system, $O_{e}X_{e}Y_{e}Z_{e}$, the roadheader coordinate system, $O_{r}X_{r}Y_{r}Z_{r}$, the tunnel coordinate system, $O_{l}X_{l}Y_{l}Z_{l}$, and the total station coordinate system, $O_{o}X_{o}Y_{o}Z_{o}$.

The global coordinate system, $O_{e}X_{e}Y_{e}Z_{e}$, has its origin at a point whose coordinates are measured with axis $X_{e}$ pointing east, axis $Y_{e}$ pointing north, and axis $Z_{e}$ perpendicular to $X_{e}$ and $Y_{e}$ and pointing upward. The roadheader coordinate system, $O_{r}X_{r}Y_{r}Z_{r}$, has the origin, $O_{r}$, located at the horizontal, rot $a_{q}$ and $a_{f}$ are, respectively, the azimuth and inclination angles, and L is the distance from the total station origin to the point to be measured.

The pose angles of the total station, output by the north-seeking gyroscope, are employed to calculate the total station coordinate system. Combining the track-ball coordinates and the coordinate system, the track-ball global coordinates can be calculated according to Equation (1).
(1)$$\left(x_{eb},y_{eb},z_{eb}\right)=(x_{ob}\times o_{x}+y_{ob}\times o_{y}+z_{ob}\times o_{z})+\left(x_{o},y_{o},z_{o}\right)$$
where $o_{x},o_{y},o_{z}$ are the normalized axis of the coordinate system, $O_{r}X_{r}Y_{r}Z_{r}$, $\left(x_{ob},y_{ob},z_{ob}\right)$ are the coordinates of the track-ball with respect to the origin, $\left(x_{o},y_{o},z_{o}\right)$, and $\left(x_{eb},y_{eb},z_{eb}\right)$ are the coordinates of the track-ball in the global coordinate system.

To position a point on the roadheader, besides the track-ball coordinates already obtained, the relative coordinates of some roadheader points relating to the track-ball coordinates and the roadheader coordinate system are required. The former can be easily measured manually while the latter can be computed as mentioned above from the pose angles from the inertial system. With the target-ball coordinates, the roadheader coordinate system, and the relative coordinates certain points can be positioned in the global coordinate system according to Equation (2). Thus, roadheader positioning can be accomplished.
(2)$$(x_{r},y_{r},z_{r})=(\Delta x_{r}\times r_{x}+\Delta y_{r}\times r_{y}+\Delta z_{r}\times r_{z})+\left(x_{eb},y_{eb},z_{eb}\right)$$
where $\left(r_{x},r_{y},r_{z}\right)$ is the normalized vector of the coordinate system, $\left(\Delta x_{r},\Delta y_{r},\Delta z_{r}\right)$ are the relative coordinates with respect to the coordinate system origin under the coordinate system $O_{r}X_{r}Y_{r}Z_{r}$, and $\left(x_{eb},y_{eb},z_{eb}\right)$ are the coordinates of the target-ball.

Cutter positioning refers to locating the cutter in the global coordinate system. Prior to locating the cutter in the global coordinate system, the relative position of the cutter in the roadheader coordinate system should be calculated. The relative position computing principle is explained in Figure 3 and Figure 4.

Figure 3 illustrates the cutter motion in a horizontal direction. ∠*MOE* equals ∠*NOF*, denoted as θ, which remains unchanged and can be measured manually as can the length of *OF* and *OE*. *M*, *M*′, *N*, and *N*′ are on the outer circle, centered at the horizontal rotation center, *O*, *R*, and *R*′ are on the inner circle, and *P*′ and *P* denote the cutter position projection on the horizontal plane at different positions. The radius of the outer and inner circles can also be measured manually. The length of *R*′*P* (*RP* in Figure 4) can be calculated according to Equation (7). Driven by the hydraulic thrust, *N* and *M* move to *N*′ and *M*′, respectively, and the cutter moves from *P*′ to *P*. The length of *EM*′ and *FN*′ can be measured by sensors equipped on the hydraulic cylinder for horizontal motion. Therefore, the angle, β, can be computed in ∆*OFN*′, according to Equation (3), and the coordinates *x* and *y* of the cutter, *P*, can be calculated following Equation (4).
(3)$$\beta =\arccos(\frac{ON^{\prime }*ON^{\prime }+OF*OF-FN^{\prime }*FN^{\prime }}{2*ON^{\prime }*OF})-\theta$$
(4)$$\{\begin{array}{c} x=(OR^{\prime }+R^{\prime }P)\sin\beta \\ y=(OR^{\prime }+R^{\prime }P)\cos\beta \end{array}$$
where $x,y,OR^{\prime },R^{\prime }P,\beta$ are showed in Figure 3.

Figure 4 illustrates the cutter motion in the vertical direction. *O*, *R*, and *P* represent the same point as Figure 3. *R* is the vertical rotation center, *A* is a fixed point, and the length of *RA* can be measured manually. ∆*RB*″*C*″ rotates around *R* in the vertical direction, and the point *C* may arrive at different positions to C, C′, and C”, of which C′ is collinear with OR. The angle ∠*ARC*′, denoted as Ω, can be measured manually. The angles ∠*B*″*R**C*″ and ∠*CRC*′, and the length of *RC*″, denoted as a, φ, and *l*, respectively, can be measured manually. In ∆*RBA*, the lengths of RA and RB are constant and can be measured manually, and the length of *AB* is measured by the sensors equipped on the hydraulic cylinder for the vertical motion of the cutter, so the angle ∠*BRA* can be computed according to Equation (5), and a can be computed according to Equation (6). Therefore, the coordinate, *z*, can be computed according to Equation (8).
(5)∠BRA=arccos((RA×RA+RB×RB−AB×AB)/(2×RA×RB))
(6)a=∠BRA−Ω+φ
(7)RP=l×cos(a)
(8)z=l×sin(a)
where ∠BRA,RA,RB,AB,RP are showed in Figure 4.

With the relative coordinates (x,y,z) of the cutter, the origin O$\left(x_{r},y_{r},z_{r}\right)$, and the coordinate system $\left(r_{x,}r_{y},r_{z}\right)$ obtained, the cutter position $\left(x_{c},y_{c},z_{c}\right)$ in the global coordinate system can be calculated in terms of Equation (9).
(9)$$(x_{c},y_{c},z_{c})=(x\times r_{x}+y\times r_{y}+z\times r_{z})+\left(x_{r},y_{r},z_{r}\right)$$

#### 2.1.2. Cutter Control Model

To carry out the cutting operation to a certain position, the coordinates of the position should be converted to the length of the hydraulic cylinder to guide the cutter moving. Consequently, the length of the hydraulic cylinder should be computed for the given position.

From Figure 4 and Equations (5)–(8), the formula to compute the length of the hydraulic cylinder for vertical motion can be obtained (Equations (10)–(12)):(10)a=arcsin(z/l)
(11)∠BRA=Ω−φ+a
(12)$$AB=\sqrt{RA\times RA+RB\times RB\times 2\times RA\times RB\times \cos∠BRA}$$
where ∠BRA,RA,RB,AB,RP are showed in Figure 4.

According to Figure 3 and Equations (3), (4), (7), and (12), the formula to compute the length of the hydraulic cylinder for horizontal cutter motion can be obtained from:(13)$$\beta =\arcsin\left(x/(OR^{\prime }+l\times \cos(a))\right)$$
(14)$$FN^{\prime }=\sqrt{ON^{\prime }\times ON^{\prime }+OF\times OF-2\times ON^{\prime }\times OF\times \cos(\beta +\theta )}$$
where $x,y,OR^{\prime },R^{\prime }P,\beta$ are showed in Figure 3. With the calculated hydraulic cylinder length, the roadheader can move the cutter to the target position. It should be noted that the given global coordinates should be converted to the relative coordinates under the roadheader coordinate system as aforementioned, and the hydraulic cylinder length should be a reachable value in a range from the minimum to the maximum value.

#### 2.1.3. Moving the Explosion Proof Positioning Box

After an advancement of about 20 m, the positioning box should move forward. The reason for this is that the total station is subject to occlusion at a too long distance. When moving the positioning box, the total station origin is changed, and it needs to be repositioned. To position the new origin, a point $\left(x_{r},y_{r},z_{r}\right)$ should be measured as a reference point and recorded before it is moved. The new origin $\left(x_{on},y_{on},z_{on}\right)$ can be calculated according to the following equation:(15)$$\left(x_{on},y_{on},z_{on}\right)=\left(x_{r},y_{r},z_{r}\right)-(x_{ob}\times o_{x}+y_{ob}\times o_{y}+z_{ob}\times o_{z})$$
where $\left(x_{ob},y_{ob},z_{ob}\right)$ is the calculated relative coordinates relating to the new origin, and $\left(o_{x},o_{y},o_{z}\right)$ are the normalized axis of the total station coordinate system.

### 2.2. Arbitrary Cross Section Trajectory Planning and Automatic Cutting

#### 2.2.1. Trajectory Planning

The cutter trajectory planning problem is actually a kind of complete coverage path problem [10], which can be solved by finding a path for the conical shaped cutter to move across the entire area of the cross section. Supposing that the radius of the cutter is *r*, the distance between any two adjacent points of the path can be chosen as *r*. The path can be designed using three methods.

The first is an S-shape (Figure 5) based method, which starts at the bottom-left corner (the circle filled with black color), then moves horizontally from left to right, then a distance of 2*r* vertically. It then moves from right to left, and finally up so that a complete loop is achieved. If such loops are repeated until the entire area is covered, the cutter trajectory is formulated. An array of three-dimensional coordinates is employed for the storage of such a trajectory. Figure 5 depicts this trajectory generation process. Furthermore, in addition to this S-shape path array, another array is required for storage of the path for swiping the border, which is traveled after the S-shape path is finished. The process for swiping borders is described in Figure 6.

The trajectory planning algorithm can be summarized as Algorithm 1.
**Algorithm 1** Trajectory planning1:*R*←the interval between the scan lines for cutter following2:*V*1←the left vertex of the bottom edge of the section3:*V*2←the right vertex of the bottom edge of the section4:for each scan line V1V2←$l_{bottom}$ to $l_{up}$5:{6: *V*1←*V*1 + $R\cdot \Delta V_{l}$// $\Delta V_{l}$ denotes the unit vector of the left directional ledge of the section7: *V*2←*V*2 + $R\cdot \Delta V_{r}$// $\Delta V_{r}$ denotes the unit vector of the ridge directional ledge of the section8: *V*←*V*19: while *V*
≠
*V*210:  {11:   *V*←*V* + $R\cdot \Delta V_{12}$// $\Delta V_{12}$ denotes the unit vector of *V*1*V*212:   Store *V*13:  }14: }15: for each e in edges{ }// edges {} refers to the bottom edges set16: {17: Compute the *V_b_*// *V_b_* refers to the border points18:   Store *V_b_*19:   }

When implementing this algorithm, supposing that the cross section coordinates are in a coordinate system with the origin at the center, the *y* axis is perpendicular to the cross section and pointing in the advancing direction, the *x* axis is pointing in the right direction, and the *z* axis is pointing in the upward direction, the vertexes of the cross section can be easily obtained with the known parameters such as width, height, and so on.

The parametric equation for each edge of the polygon can be obtained according to its vertexes, which can be used to discretize the edge to obtain the arrays of points to act as the end points of each scan line segment. It should be noted that the points of the possible horizontal edge of the polygon can be represented with only the two end points, as Equation (16).
(16)$$\left(x_{c},y_{c},z_{c}\right)=\left(x_{p},y_{p},z_{p}\right)+\Delta \times (x_{d},y_{d},z_{d})$$
where $\left(x_{c},y_{c},z_{c}\right)$ represents the current point, $\left(x_{p},y_{p},z_{p}\right)$ represents the previous point already calculated, and Δ represents the length increment in the direction $\left(x_{d},y_{d},z_{d}\right)$.

With the generated edge point array, each horizontal scanning line segment can be obtained, which will be taken as one part of the whole trajectory generation. When generating the trajectory, as aforementioned, the bottom up order is followed.

It should be noted that the trajectory point array should be converted to the global coordinate system according to the aforementioned method, which will be further converted to the coordinates under the roadheader coordinate system for guiding the roadheader cutter movement.

The second method is based on a decreasing cross section polygon area size. The algorithm can be described as follows.

This method first discretizes the part of the polygon area that is bound by the original edges of the polygon and their parallel line segments in the inner part of the polygons, the distance between which is set as the diameter of the cutter. Then, taking the parallel inner line segments as the border, a smaller polygon is formulated, which can be discretized in the same way as the original polygon discretization. By repeating this operation until all of the polygon area is discretized, the trajectory can be obtained.

The third method can be employed for arbitrary trajectory generation, which allows the operator to generate the trajectory. This method can be described as follows.

The cross section should be decomposed into a collection of uniform grid cells of *n* rows and *m* columns, which is stored as a two-dimensional array of a size *n* by *m*.

The area of the screen should be divided into a collection of uniform grid cells in the same way, which should be mapped exactly with the cross section grids.

The position where the mouse is pressed down should be picked and converted into the row and the column of the screen grids, which will be employed to calculate the exact grid of the cross section. Once the trajectory point selections by the mouse are accomplished, the trajectory is generated.

#### 2.2.2. Cutting Process

According to Figure 5 and Figure 6, prior to use for cutting the cross section, the cutter trajectory should be planned. A point array will be generated that stores the position coordinates for guiding the motion of the cutter. Then, the points will be sent one by one, followed by the hydraulic cylinder length computing for the sent points. The length will be delivered to the programming logic control unit (PLC) for guiding the cutting head motion. Meanwhile, the cutter head position will be calculated and compared with the target point. If the target point is reached, the next point will become the new target point. When all the points in the array have been traversed in order, the cutting process is finished. If both the trajectory path and the path for swiping the border are accomplished, the cutting process will be stopped.

## 3. Experiments and Results

Experiments were carried out underground on an EBZ160 roadheader for coal mining at the Zhaizhen Mine, which is owned by Xinwen Mining Group Co., Ltd. (Tai’an, China). The tests of the experiments can be divided into two steps, of which the former is the target-ball positioning test, that is, the roadheader positioning test, and the latter is the cutter positioning test. To obtain the actual results, another total station, named as the test total station, with the direction labeled strictly in line with the north direction, was employed as the global positioning system to measure the results directly. The total station for the roadheader positioning model was named as the measure total station. The results from the two total stations were compared to test the error. Before the experiments began, the test total station was situated so that the two total stations were under the same coordinate system. The test total station was employed to measure two points, which were utilized for the positioning of the measure total station. With the positioned measure total station, the positioning results of the target-ball were output by the positioning program, which were obtained by combining the outputs of both the positioning total station and the pose measurement of the total station by the north-seeking compass.

These results are shown in Table 1. In Table 1, Tball_*mx*, Tball_*my*, and Tball_*mz* are the coordinates of the target-ball from the measure total station, while Tball_*x*, Tball_*y*, and Tball_*z* are the coordinates from the test total station. Root mean square error (RMSE) provides an accuracy index at a global level, which is expressed as follows:(17)$$RMSE=\sqrt{\frac{\sum_{i=1}^{n}{\left(m_{i}-p_{i}\right)}^{2}}{n}})$$
where $m_{i}$, $p_{i}$, and n denote the *i*th result by the test total station, the *i*th positioning result by the measure total station, and the number of the validation points, respectively. RMSE was employed to estimate the error of the positioning results. The errors of each point are showed in Table 2. RMSE for *x*, *y*, and *z* were calculated as 0.049602, 0.02055, and 0.027588, respectively, with meter (m) as the unit.

The origin of the global coordinate system was set at the point (42.991, −11.74, 3.294), the pose angles yaw, pitch, and roll from the gyroscope, denoted as α, β, and γ were, respectively, 292.495, 0.0055, and −1.1044, and the pose angles α, β, and γ from the inertial system were 65.72, −0.093, and 0.083, respectively. With the pose angles from the gyroscope, the target-ball coordinates were computed, and the results were 15.113, −1.648, and 4.08. With the relative coordinates (−0.458, 2.865, and −0.704) and the pose angles from the inertial system, the roadheader center coordinates were computed as 12.69037, −0.0538364, and 3.370688. Supposing these were set before the cutting head starts to work, with different real-time lengths of the hydraulic cylinder, different cutter point coordinates could be computed. To test the error of the cutter positioning, a cross section was built according to the aforementioned cutting process. This cross section was of a size of 5.2 by 4.5 m. According to Figure 7, the cross section was discretized in steps of 0.8 m, and the trajectory was generated and stored in a point array with a size of 120. The cutter moved from the first point at the bottom left position to the center point according to the trajectory, and Figure 8 shows the cutting process for the cross section. After the cutting process was completed, the cross section was formulated, as showed in Figure 8.

The results from the cutter positioning for the whole cross section were calculated, and only the corner points are showed in Table 3, since only the border has an impact on the accuracy of the cross section shape formulation. Table 3 shows the coordinate comparisons, and the errors of each point coordinate are shown in Table 4. RMSE values for *x*, *y*, and *z* were calculated, and the results were 0.059937113, 0.013302904, and 0.01617986, respectively. For the whole cross section, the errors for 120 sampled points were calculated and yielded RMSE value of 0.041254321, 0.012413522, and 0.01306654.

The other two trajectory generation methods were also tested and employed for automatic cutting (Figure 9 and Figure 10). Figure 9 shows the automatic cutting process according to the S-shaped trajectory planning, while Figure 10 shows the automatic cutting process according to the arbitrary trajectory planning.

## 4. Discussion

Positioning models for the roadheader and the cutter were proposed, and an automatic cutting process based on the positioning models was also proposed. The experiments for the positioning models were carried out and automatic cutting of a practical tunnel was tested according to the automatic cutting method.

According to the experimental results for roadheader positioning, it was verified that the roadheader positioning model could achieve a result with a high accuracy when positioning the roadheader. The maximum error was less than 5 cm, which satisfies the requirements of roadheader positioning and automatic cutting. The reason for this can be attributed to the adoption of high accuracy total station, which provides an accurate result. Furthermore, it should be noted that the most important factor is the adoption of a gyroscope and the positioning computing algorithm fused with the pose angles information of total station, which efficiently and accurately generate the precise positioning result.

As for the cutter positioning, the error that is obtained is also a small value, which results in a high accuracy. This result is achieved from the cutter positioning model, which is impacted by both the cutter relative positioning method and the cutter positioning model. It should also be noted that the parameters of the roadheader determine the cutter relative positioning model. Only with the optimized parameters can the cutter relative positioning obtain a result with a high accuracy. It should be noted that parameter optimization is a non-trivial work that requires great patience. Additionally, the precise pose angles contribute greatly to the accurate positioning of the cutter.

Three trajectory generation methods are provided, which can support the trajectory planner in a very flexible way. With the three methods for trajectory generation, the automatic cutting process can be well carried out, and the shape of the cross section can be well formulated, which lays the foundation for complete automatic cutting with automatic roadheader movement.

Due to the accurate positioning of the roadheader and cutter, the automatic cutting of practical tunnels exhibits a high accuracy, which achieves a result with an error of 5 cm. It should be noted that, although the cross section shape is well formulated, the errors of the corner points and the whole cross section points differ. Furthermore, the corner points have a bigger error than the whole of the cross section points. This may indicate that the quality of the formulated shape is impacted not only by the positioning model but also by the speed change of the cutter, and the latter may have an obvious negative impact on the cutting error, which will be further studied.

## 5. Conclusions

In this paper, to address the roadheader and the cutter positioning problem in underground environments, by combining multiple sensors, such as the inertial navigation system, total station, and a north-seeking gyroscope, a positioning method is proposed, and a corresponding positioning model is built. The experiment demonstrates that the positioning method can realize the precise positioning of both the roadheader and the cutter. Furthermore, with the flexible trajectory planning methods provided, and based on the positioning method, an automatic cutting process is proposed, and practical automatic tunneling exhibits a well formulated shape and high formulation accuracy. The positioning model and automatic cutting process lays a sound foundation for smart tunneling, and the automatic cutting process requires no intervention from the operators and can be utilized to implement complete autonomous tunneling, which will be the future focus of our work.

## Figures and Tables

**Figure 1 sensors-19-04955-f001:**
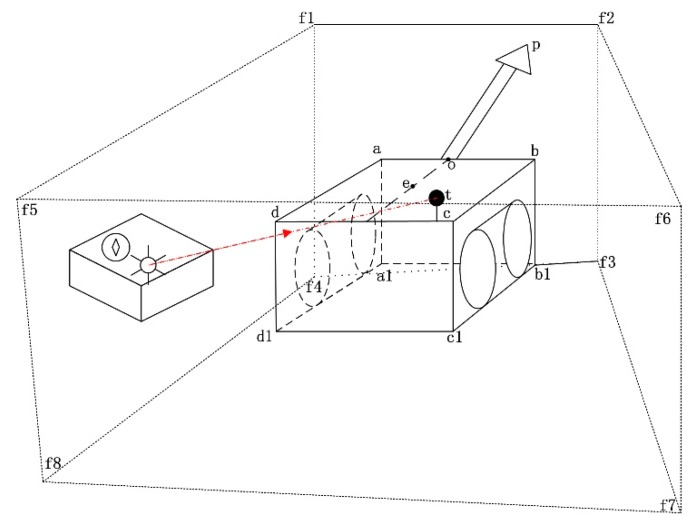
The principle of robotic roadheader positioning. The cube with the vertices *f*1, *f*2, *f*3, *f*4, *f*5, *f*6, *f*7, and *f*8 represents the tunnel, the cube with the vertices *a*, *b*, *c*, *d*, *a*1, *b*1, *c*1, and *d*1 represents the roadheader body, *p* is the cutter, and *t* is the target-ball, which receives the laser from the positioning box on the left.

**Figure 2 sensors-19-04955-f002:**
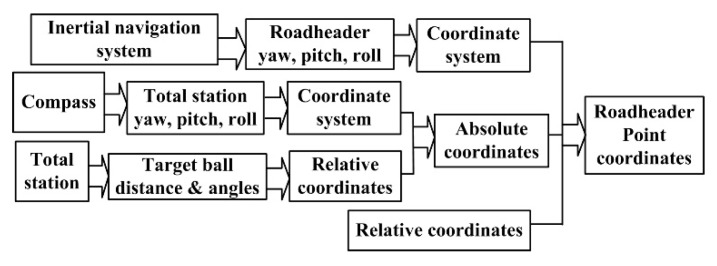
Positioning workflow.

**Figure 3 sensors-19-04955-f003:**
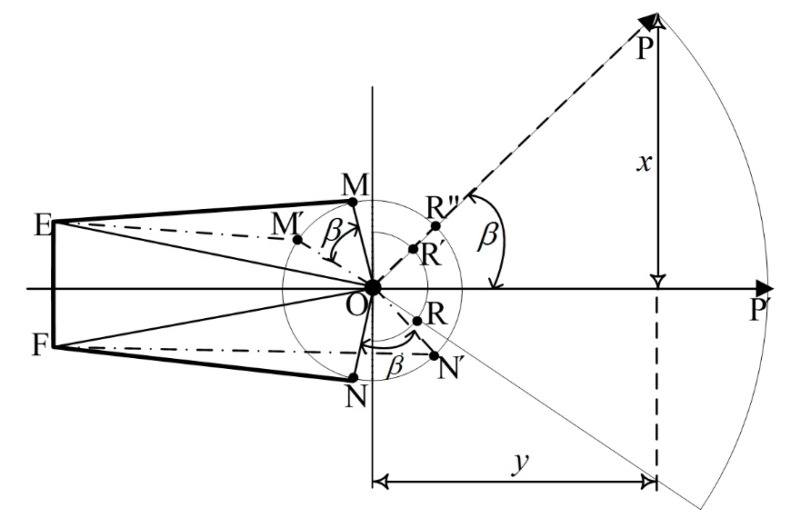
Cutter motion in horizontal direction. *M*, *M*′, *N*, and *N*′ are on the outer circle, centered at the horizontal rotation center, *O*, *R*, and *R*′ are on the inner circle, and *P*′ and *P* denote the cutter position projection on the horizontal plane at different positions.

**Figure 4 sensors-19-04955-f004:**
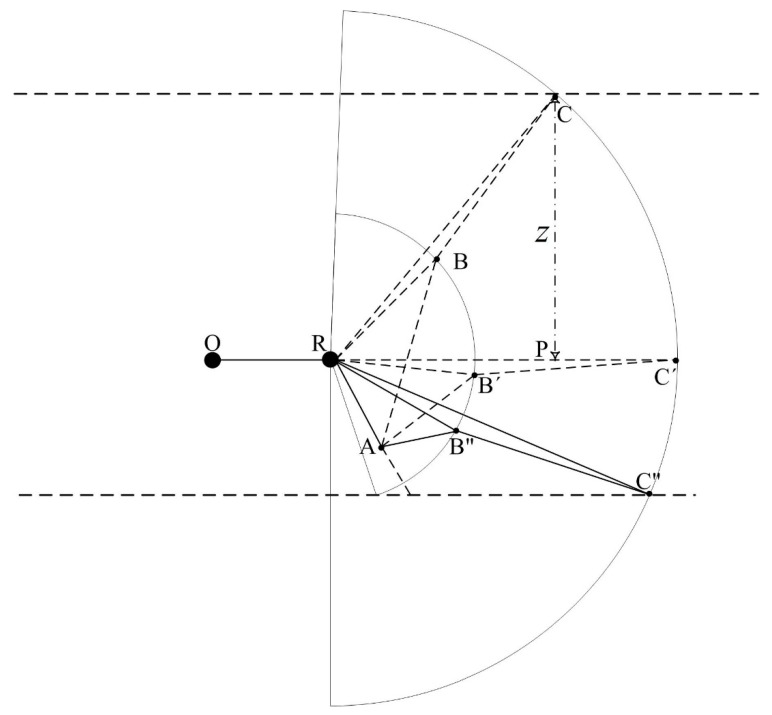
Cutter motion in the vertical direction.

**Figure 5 sensors-19-04955-f005:**
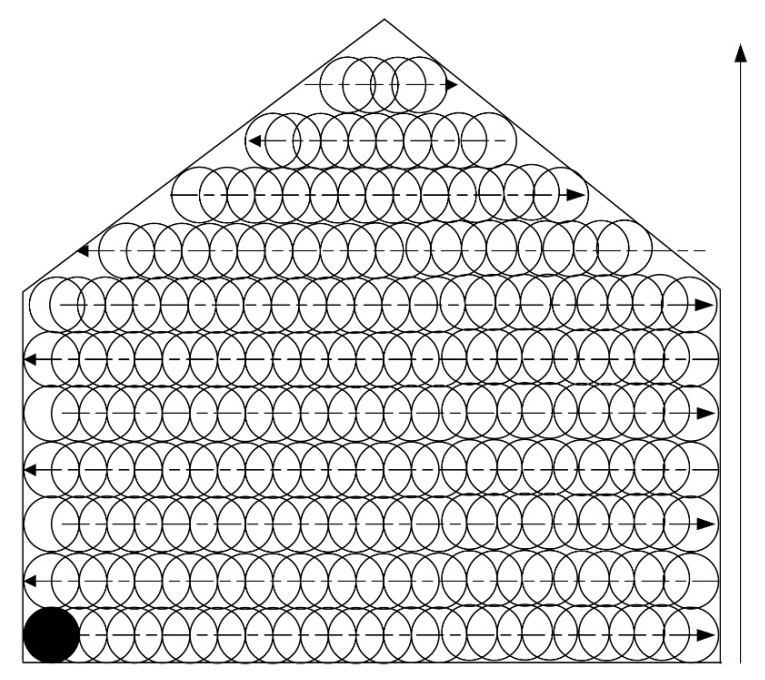
Trajectory planning for inner cross section.

**Figure 6 sensors-19-04955-f006:**
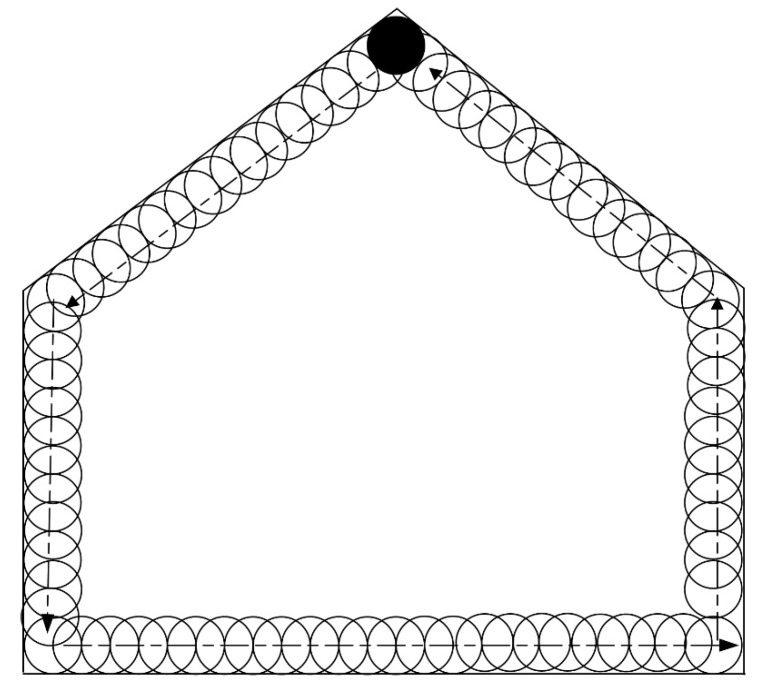
Trajectory planning for border swiping.

**Figure 7 sensors-19-04955-f007:**
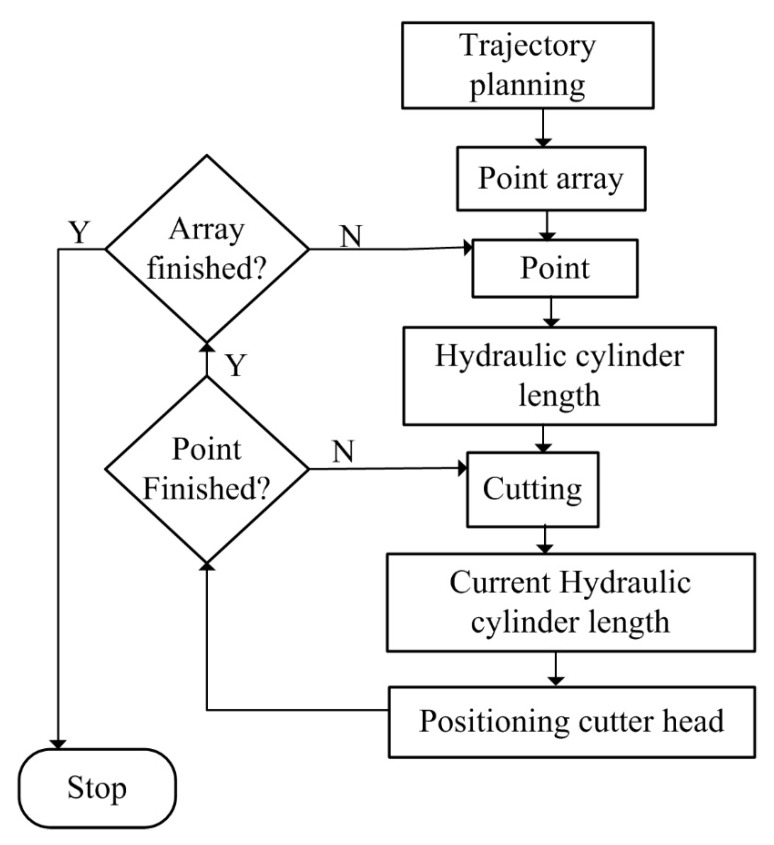
Workflow for robotic roadheader.

**Figure 8 sensors-19-04955-f008:**
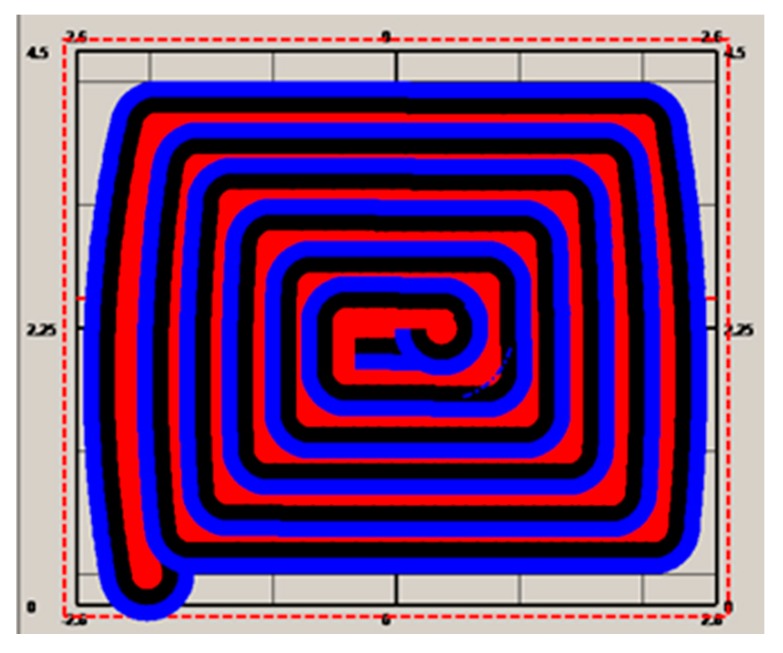
Decreasing polygon size trajectory planning and the automatic cutting process by the robotic roadheader.

**Figure 9 sensors-19-04955-f009:**
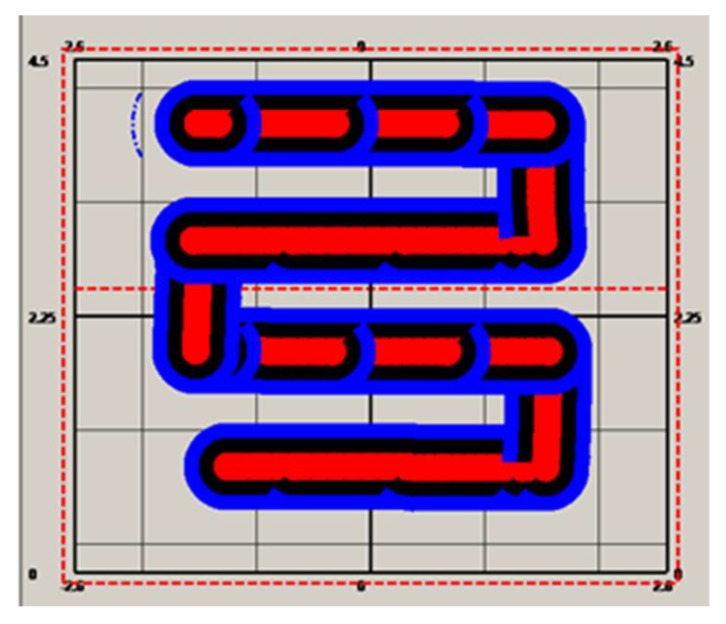
S-shaped trajectory planning and cutting.

**Figure 10 sensors-19-04955-f010:**
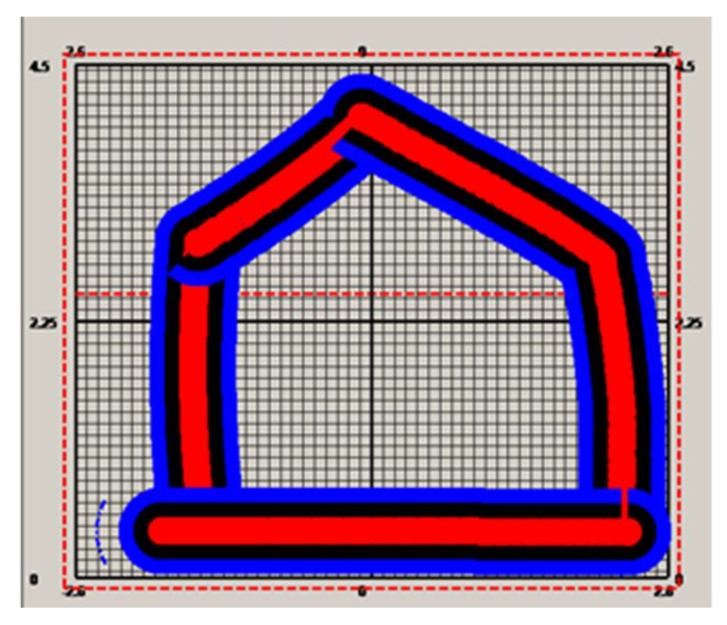
Arbitrary trajectory planning and cutting.

**Table 1 sensors-19-04955-t001:** Comparison between the real coordinates and the positioning coordinates (m). Tball_*mx*, Tball_*my*, and Tball_*mz* are the coordinates of the target-ball from the measure total station, while Tball_*x*, Tball_*y*, and Tball_*z* are the coordinates from the test total station.

Tball_*mx*	Tball_*my*	Tball_*mz*	Tball_*x*	Tball_*y*	Tball_*z*
2.551	−6.025	3.796	2.473	−6.02	3.80201
7.261	−3.657	3.759	7.22	−3.678	3.75789
−2.549	6.059	3.513	−2.627627	5.301803	3.49537
−2.958	6.864	6.143	−3.01761	6.247142	6.168197
−2.679	6.248	1.945	−2.724617	5.516208	1.926952
−5.566	5.526	2.959	−5.802043	4.944234	2.934521
−2.639	6.146	3.44	−2.713292	5.431437	3.5042
−5.627	5.637	3.99	−5.81226	4.942852	3.99768

**Table 2 sensors-19-04955-t002:** Roadheader positioning error (m).

Error_*x*	Error_*y*	Error_*z*
0.07174	−0.00259	−0.00601
0.03161	0.020911	0.00111
0.078627	0.004197	0.01763
0.05961	0.016858	−0.0252
0.045617	0.011792	0.018048
0.036043	−0.01823	0.024479
0.014292	0.014563	−0.0642
−0.01474	0.044148	−0.00768

**Table 3 sensors-19-04955-t003:** Comparison between the real coordinates and the positioning coordinates (m).

Cutter_*mx*	Cutter_*my*	Cutter_*mz*	Cutter_*x*	Cutter_*y*	Cutter_*z*
2.117	8.186	4.597	2.145352	8.158442	4.571989
2.117	8.129	2.423	2.166604	8.106151	2.392711
1.929	8.468	5.442	1.99126	8.485817	5.47986
−0.232	7.545	3.521	−0.16831	7.49267	3.503577
3.383	8.803	3.547	3.386812	8.792561	3.496827

**Table 4 sensors-19-04955-t004:** Error table for the cutter positioning (m).

Error_*x*	Error_*y*	Error_*z*
0.07174	−0.00259	−0.00601
0.03161	0.020911	0.00111
0.078627	0.004197	0.01763
0.05961	0.016858	−0.0252
0.045617	0.011792	0.018048

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
