# Peer review of "A Multi-Sensor Based Roadheader Positioning Model and Arbitrary Tunnel Cross Section Automatic Cutting"

_sensors, 2019, doi:10.3390/s19224955_

Round 1

Reviewer 1 Report

The paper presents a multi-sensor positioning algorithm for automated tunneling applications. Overall the paper is well-written and accompanied by experimental results from a realistic environment (coal mine). The authors should note the following minor points:

There are some typos in the paper that should be fixed. For example, line 154: "role" -> "roll". Fig. 4: Labels are very small and hard to read; please make them larger. Line 280, Alg. 1: Please write out the algorithm in standard format, e.g., using pseudocode or a flowchart, so that it is easier to read. Please include a picture of the experimental setup if possible; it would be of interest for the readers. Please specify the units used for Tables 1-4 and related text. Are the data points in meters?

Author Response

Please see the attachment for author's reply.

Reviewer 2 Report

At present, the underground high-precision positioning mainly adopts the total station method. The biggest disadvantage of this method is that it needs to be equipped with professional technicians, and the positioning effect is poor when there is dust and vibration interference. In this paper, a fusion method of multi-sensor for road-header position is proposed and tested. At present, relevant products are available. I think this way of positioning will be widely used in engineering practice. I think this study has a great important theoretical significance and engineering application value

But as for this manuscript, I have the following questions and suggestions,

(1) Is this method first proposed by the author?
(2) It is suggested to make a comprehensive review of existing products and literatures, and analyze their advantages and disadvantages to reflect the contribution of this study
(3) How is the accuracy and superiority of the scheme proposed by the author compared with other schemes? It is suggested that the advantages of this study should be reflected from the perspective of performance parameters.

In order to improve the quality of the manuscript, I suggest that you revise it and resubmit it.

Author Response

Please see the attachment for the author's reply.

Round 2

Reviewer 2 Report

In view of the author's clarification of my doubt, I agree to accept this manuscript. However, in order to improve the manuscript, it is still recommended to check the full text carefully.